# SARS-CoV-2 Infection during Delivery Causes Histopathological Changes in the Placenta

**DOI:** 10.3390/diseases12070142

**Published:** 2024-07-02

**Authors:** Jędrzej Borowczak, Agnieszka Gąsiorek-Kwiatkowska, Krzysztof Szczerbowski, Mateusz Maniewski, Marek Zdrenka, Marta Szadurska-Noga, Karol Gostomczyk, Paula Rutkiewicz, Katarzyna Olejnik, Wojciech Cnota, Magdalena Karpów-Greiner, Wojciech Knypiński, Marta Sekielska-Domanowska, Grzegorz Ludwikowski, Mariusz Dubiel, Łukasz Szylberg, Magdalena Bodnar

**Affiliations:** 1Department of Tumor Pathology and Pathomorphology, Oncology Centre-Prof. Franciszek Łukaszczyk Memorial Hospital, 85-796 Bydgoszcz, Polandkarolgostomczyk.research@gmail.com (K.G.);; 2Department of Obstetrics, Gynaecology and Oncology, Collegium Medicum in Bydgoszcz, Nicolaus Copernicus University in Torun, 85-168 Bydgoszcz, Polandmagdalenakarpow@wp.pl (M.K.-G.); wojciech.knypinski@gmail.com (W.K.); marta.sekielska@cm.umk.pl (M.S.-D.); dubiel@cm.umk.pl (M.D.); 3Doctoral School of Medical and Health Sciences, Nicolaus Copernicus University in Torun, 85-094 Bydgoszcz, Poland; 4Department of Pathomorphology and Forensic Medicine, Faculty of Medical Sciences, University of Warmia and Mazury, 10-561 Olsztyn, Poland; szadurskamarta@gmail.com; 5Chair of Pathology, Jan Biziel University Hospital No. 2, 85-168 Bydgoszcz, Polandolejnikkasia05@gmail.com (K.O.); 6Chair and Department of Gynaecology and Obstetrics, Faculty of Health Sciences in Katowice, Medical University of Silesia, 41-703 Ruda Śląska, Poland

**Keywords:** COVID-19, placentitis, placenta, histopathology, complications

## Abstract

Background: SARS-CoV-2 can damage human placentas, leading to pregnancy complications, such as preeclampsia and premature birth. This study investigates the histopathological changes found in COVID-19-affected placentas. Materials and Methods: This study included 23 placentas from patients with active COVID-19 during delivery and 22 samples from patients without COVID-19 infection in their medical history. The samples underwent histopathological examination for pathology, such as trophoblast necrosis, signs of vessel damage, or fetal vascular malperfusion. Results: Newborns from the research group have lower weights and Apgar scores than healthy newborns. In the COVID-19 group, calcifications and collapsed intervillous space were more frequent, and inflammation was more severe than in the healthy group. At the same time, the placenta of SARS-CoV-2-positive patients showed signs of accelerated vascular maturation. Trophoblast necrosis was found only in the placentas of the research group. The expression of CD68+ was elevated in the COVID-19 cohort, suggesting that macrophages constituted a significant part of the inflammatory infiltrate. The increase in lymphocyte B markers was associated with placental infarctions, while high levels of CD3+, specific for cytotoxic T lymphocytes, correlated with vascular injury. Conclusions: SARS-CoV-2 is associated with pathological changes in the placenta, including trophoblast necrosis, calcification, and accelerated villous maturation. Those changes appear to be driven by T cells and macrophages, whose increased expression reflects ongoing histiocytic intervillositis in the placenta.

## 1. Introduction

Coronavirus disease 2019 (COVID-19) is a highly contagious viral disease caused by severe acute respiratory syndrome coronavirus 2 (SARS-CoV-2) [1]. Since the outbreak of the pandemic, it caused over 6 million deaths and paralyzed almost every area of human life [2]. SARS-CoV-2 primarily attacks the respiratory system, causing a broad spectrum of symptoms from mild upper respiratory tract ailments to pneumonia, acute respiratory distress syndrome, and even death. Other symptoms, such as fatigue, infertility, or septic shock, may also occur [2,3,4]. Due to impaired immune response, patients with immunodeficiency and underlying comorbidities appear more prone to complications from the infection [5]. Despite great progress in recent years, many COVID-19 complications are still unknown [6].

SARS-CoV-2, like other coronaviruses, is a positive-stranded, enveloped RNA virus [3]. Besides RNA, the virus is built from structural proteins, including spike, membrane, nucleocapsid, and envelope proteins, alongside nonstructural proteins, responsible for its unique abilities [7]. The angiotensin-converting enzyme 2 (ACE2) receptors constitute the main way SARS-CoV-2 enters human cells. Upon its binding by the viral spike proteins, SARS-CoV-2 hijacks host cell proteases, leading to the release of viral genetic material into the host cell [8]. Since tissue damage is related to how SARS-CoV-2 enters host cells, the pattern of ACE2 receptor expression seems to explain the variety of symptoms associated with COVID-19 [9]. The concentration of ACE2 receptors is the highest in the pulmonary system, making it the most vulnerable to infection [10]. Nevertheless, ACE2 receptors are also present in other tissues, such as the gastrointestinal tract and placenta, predisposing their injury during infection [9,11].

Pregnancy is associated with multiple physiological changes in the female body, which are not inert for the organism and may lead to a wavering of homeostasis, especially in the presence of pathogens [12,13,14]. Due to physiological changes in adaptive immunity, pregnant women are prone to SARS-CoV-2 injections and fall ill more often than the general population [15]. As the pregnancy progresses, the suppression of the immune system increases to allow the growth of a semi-allogeneic fetus [16]. Therefore, the immune response to infections is more robust in pregnancy’s first and third trimesters. This is likely due to the proinflammatory state of the placenta, which supports trophoblast development in the first trimester and prepares the fetus for delivery in the third trimester [17]. Clinical complications also differ between trimesters, with an increased risk of miscarriage in the first trimester and more frequent preterm deliveries in the third trimester [18,19]. Accordingly, only 3% of pregnant women are infected with SARS-CoV-2 during the first trimester, which rises to 83.58% during the third trimester [20]. This association may be related to the breakdown of the placental tissue barrier caused by proinflammatory signaling [17].

COVID-19 can lead to increased pregnancy complications, such as maternal vascular malperfusion, preeclampsia, premature birth, and growth retardation [13,21,22,23]. Furthermore, even asymptomatic infections can cause premature aging and impaired uterine blood flow in the placenta, disrupting fetal growth [24,25]. The mechanism of SARS-CoV-2-related complications seems to involve placenta injury that derives from virus–host interactions. Multiple pathomorphological changes in the placenta are also associated with SARS-CoV-2 infection, including increased fibrin deposition, trophoblast necrosis, and intervillous thrombi [23,26,27]. Thus, it is critical to determine the associations between maternal SARS-CoV-2 infection, virus–host interaction, and placenta pathology to assess its possible impact on pregnancy complications. Although recent studies suggest that SARS-CoV-2 vertical transmission is rare, COVID-19 can influence pregnancy through placenta-related mechanisms [26,28]. Therefore, understanding the pathomechanism of SARS-CoV-2 infection is critical for improving patient care.

This study examined damage-related pathomorphological features in human placentas, including maternal and fetal vascular malperfusion, delayed vascular maturation, signs of intrauterine infection, and villitis of unknown etiology, as well as other pathologies found in COVID-affected placentas. All samples were also stained for markers found on inflammatory cells to determine the nature of those findings.

## 2. Materials and Methods

### 2.1. Patients and Tissue Samples

The research group includes 23 placentas from patients with active COVID-19 during delivery, and the control group consisted of 22 placentas from patients without COVID-19. All patients in this study were randomly recruited during admission to the Department of Perinatology, Obstetrics, and Oncology between November 2020 and May 2021 in Bydgoszcz (Poland). All tissue specimens were collected immediately after delivery. We collected the available clinical data, including the gestational ages of 18 patients from the research group and 22 patients from the control group. Five patients with confirmed COVID-19 refused the collection of anonymous clinical data. Patients with a medical history of COVID-19 during pregnancy were excluded from this study. Due to the possible impact of SARS-CoV-2 infection on delivery time, the cohorts were not matched for similar gestation ages. Clinical data of the COVID-19 group neonates, including body weight and APGAR score, were also collected. This study followed the Declaration of Helsinki, and the protocol was approved by the Bioethics Committee (KB 619/2021).

### 2.2. Confirmation of the SARS-CoV-2 Infection

The rapid antigen test initially examined all patients for SARS-CoV-2 during admission to the hospital. All results were confirmed by the RT-PCR test before delivery. A molecular test on nasopharyngeal swab was performed for all newborns of the mothers from the COVID-19 group within 48 h after delivery. Viral RNA was isolated using the Viral RNA/DNA Purification Kit (EURx Sp.z o.o., Gdańsk, Poland). The deparaffinization step was added to the manufacturer’s original protocol to allow the extraction of nucleic acids from formalin-fixed, paraffin-embedded (FFPE) tissue and followed the standard protocol described in the literature [29]. The detection of SARS-CoV-2 virus RNA was performed on the CFX96 Real-Time PCR Detection System (Bio-Rad Inc., Hercules, CA, USA) using the SARS-CoV-2 Triplex PCR kit (Astra Biotech GmbH, Berlin, Germany), which simultaneously detects three targets in ORF1ab, ORF8, and N protein-coding regions of SARS-CoV-2 genome. The RNA extraction and reverse transcription reaction quality was monitored in each sample by an internal control reaction. In each run, a positive control (a mixture of in vitro transcribed RNAs corresponding to target regions of the SARS-CoV-2 genome) and a negative control reaction were performed.

### 2.3. Histochemical Staining

To visualize trophoblast necrosis, calcifications, fibrin depositions, signs of inflammation, and vessel damage, staining with classical hematoxylin and eosin and Masson trichrome (DiaPath, 010210) was performed. Selected paraffin blocks were cut using a manual rotary microtome (Leica) into 4.0 μm thick paraffin sections and placed on extra adhesive slides (SuperFrost Plus; Menzel-Glaser, Braunschweig, Germany). Subsequently, histochemical staining with H&E and Masson trichrome (DiaPath, 010210), according to the protocol recommended by the manufacturer, was performed.

### 2.4. Histopathological Evaluation

Clinical data were blinded, and the specimens were scanned using a Ventana DP200 slide scanner. Then, the scans were individually examined by an experienced pathologist to find signs of maternal vascular malperfusion (MVM), fetal vascular malperfusion (FVM), delayed villous maturation (DVM), ascending intrauterine infection, and villitis of unknown etiology (Figure 1). Sampling for histopathology was performed following the standard protocol used in the Department of Pathomorphology. In each case, a minimum of 3 placental blocks were taken, including samples of the transverse sections of the umbilical cord, roll of membranes, and full-thickness blocks of the placental parenchyma. The evaluation was performed according to the Amsterdam Placental Workshop Group Consensus Statement [30]. The placental lesions were also defined based on the Amsterdam criteria, e.g., delayed vascular maturation was defined as a monotonous, villous population with reduced numbers of vasculosyncytial membranes for the gestation period. Independently, two experienced pathologists evaluated the specimens for trophoblast necrosis, calcifications, fibrin depositions, signs of inflammation, and vessel damage. The severity of inflammation was categorized as slight (few inflammatory lymphocytes), moderate (a small group of 10–20 lymphocytes), considerable (large groups of >20 lymphocytes or multiple small groups), or severe (diffuse lymphocyte infiltration). Furthermore, some findings were classified depending on severity.

### 2.5. Immunohistochemical Staining

Paraffin blocks were sectioned into 3.0 μm paraffin sections using Leica manual rotary microtome and placed on adhesive slides (SuperFrost Plus; Menzel-Glaser, Braunschweig, Germany). Then, an automated BenchMarkGX slide processing system (Ventana Medical Systems, Tucson, AZ, USA) was used to perform the immunohistochemical (IHC) reaction. The UltraView DAB IHC Detection Kit (Ventana Medical Systems, Tucson, AZ, USA) was used for visualization, as recommended by the manufacturer. The list of used antibodies and their descriptions is provided in Table 1. All antibodies were pre-diluted and optimized for compatibility with the Ventana benchmark. IHC reactions were performed automatically following the manufacturer’s protocols (Ventana Medical Systems, Tucson, AZ, USA). Before being examined, the samples were progressively dehydrated using a series of progressive ethanol concentrations (80%, 90%, 96%, and 99.8%), cleaned in xylene (steps I to IV), and mounted with a mounting medium.

### 2.6. IHC Evaluation

Two experienced pathologists captured the images of the most representative areas of each sample using an optical microscope at x10 magnification. To minimize the risk of bias, clinical data were blinded prior to sample assessment, and immunohistochemical expression was evaluated using ImageJ 1.53j version (NIH, Bethesda, MD, USA) (Java 1.8.0_172) and the IHC profiler plugin. We followed the standard protocol designed by Verghese et al. to determine vimentin expression in the placenta [31].

IHC staining intensity of clusters of differentiation (CD3, CD4, CD8, CD20, CD68, and CD138), as well as the size of fiber depositions, the intensity of intervillositis and blood extravasations, the presence of calcifications, the collapse of intervillous space, and trophoblast necrosis, were assessed by two experienced pathologists. CD staining intensity was based on the percentage of stained cells within a representative region of each sample. We adopted the following formula: <1% of stained cells reflected a slight staining intensity (1); 1–5%, a moderate intensity (2); 5–10%, a considerable intensity (3); 10–25%, a strong intensity; and >25% of stained cells, a very strong sensitivity (5). The expression of CD34+ was measured in ImageJ by calculating the number of CD34+-positive cells in a representative area as a percentage of all cells in that area. Samples with 18.5% or more CD34-positive cells were considered as high CD34+, while samples with a percentage of CD34-positive cells below 18.5% were considered low CD34+. Vimentin expression was calculated using ImageJ. The intensity range was 61 to 120 for the positive zone, 121 to 180 for the low positive zone, and 181 to 235 for the negative zone. A vimentin H-score of 217 and higher was considered high. An H-score below 217 was considered low. The H-score was calculated using the following formula: [1 × (%cells low positive) + 2 × (%cells positive) + 3 × (%cells high positive)], with 0 being the lowest possible expression and 300 being the highest possible expression.

### 2.7. Statistical Analysis

Statistica version 13.3 (Statsoft) and Microsoft Excel 2019 were used for the statistical analyses. The normality of variables was tested using the Shapiro–Wilk test. Intergroup differences were analyzed using Pearson’s chi-squared or the U Mann–Whitney test. The Kruskal–Wallis test was employed to ascertain whether there were significant differences between more than two independent groups. Correlations between variables were evaluated using Spearman’s rank correlation coefficient for categorical data and Phi/Pearson coefficient for binary data. A *p*-value of <0.05 was deemed to be statistically significant.

## 3. Results

### 3.1. Patient Characteristics

The research group comprised 23 placentas from patients with active SARS-CoV-2 infection during delivery (Table 2). The control group included 22 placentas from patients with no SARS-CoV-2 infection medical history. The mean patient age was 31 years (range 24–38) in the COVID-19 group and 31 years (range 22–43) in the control group. The mean gestational age at the time of sample collection was 34 weeks in the COVID-19 group and 38 weeks in the control group. Newborns from the research group had lower weights (2140 g vs. 2395 g, respectively) and Apgar scores in the first minute after delivery than newborns in the control group. We also collected data from post-birth COVID-19 tests of the newborns. None of the newborns in our group were positive; 13 were negative, and 10 samples were inconclusive.

### 3.2. Histological Findings in Placentas

Initially, all samples underwent histopathological examination in search of COVID-associated changes. Five main pathologies were considered: maternal vascular malperfusion, fetal vascular malperfusion, delayed vascular maturation, signs of intrauterine infection, and villitis of unknown etiology (Table 3). Furthermore, we also assessed the presence of abundant fibrin depositions, calcification, intervillous space collapse, and trophoblast necrosis, the intensity of vascular inflammation (intervillositis), and the extravasations from blood vessels into the villous stroma (Table 4).

In the COVID-19 group, fibrin depositions, calcifications, and collapsed intervillous spaces were more numerous, while inflammation was more severe than in healthy placentas (*p* < 0.05). The presence of collapsed intervillous space, calcifications, and the intensity of intervillositis were strongly associated with active COVID-19 (k = 0.3, k = 0.51, and k = 0.57, respectively; *p* < 0.05) (Table 4). Both groups did not differ in the intensity of blood excavations.

Trophoblast necrosis was found only in the placentas of patients with active COVID-19 (20/23, *p* < 0.0001), and its presence strongly correlated with active SARS-CoV-2 infection (k = 0.88; *p* < 0.05). Signs of accelerated vascular maturation were also more common in infected placentas than in healthy ones (*p* < 0.05). While maternal vascular malperfusion, fetal vascular malperfusion, and distal villous hypoplasia were more common in COVID-19-positive placentas, the difference lacked statistical significance (*p* > 0.05).

### 3.3. The Differences in the Distribution of Assessed Protein Expression

All collected placenta specimens were stained for the expression of CD3+, CD4+, CD8+, CD20+, CD34+, CD68+, CD138+, and vimentin (Figure 2). Staining intensity was assessed on a 1–5 scale in the case of most CD markers and as high or low in the case of CD34 and vimentin. We found no statistical difference in CD3-, CD8-, and CD20-positive cell populations between examined cohorts (*p* < 0.05). CD34 expression was lower in the COVID-19 group (p = 0.048), while CD4, CD68, and vimentin levels were higher in the COVID group (*p* < 0.05) [Table 5].

### 3.4. Correlation between Histopathological Findings and Immunohistochemical Markers

Intending to explain the pathogenesis of the found changes, we evaluated the correlations between histopathological findings and immunohistochemical markers in placentas from SARS-CoV-2-positive women. We found that diffuse fibrin depositions were associated with high CD8 and CD68 expression, characteristic of cytotoxic T cells and histiocytes, respectively (*p* < 0.05). Delayed vascular maturation correlated with lower expression of CD3, a marker of T cells, while high expression of CD20, a standard B cell marker, was correlated with placental infarctions (*p* < 0.05). The presence of calcifications was associated with lower expression of CD34, present on the vascular epithelium (*p* < 0.05). The correlation between CD20 and diffuse fibrin depositions and the correlation between the intensity of intervillositis and CD68 were borderline significant (*p* = 0.513 and *p* = 0.053, respectively) [Table 6].

## 4. Discussion

### 4.1. Placenta Susceptibility to SARS-CoV-2 Infection

Placentas are prone to SARS-CoV-2 infection due to physiological immunosuppression during pregnancy and the presence of ACE2 receptors, but their susceptibility depends on their maturity and placental barrier integrity [32]. The placental barrier consists of two main layers: mononucleated cytotrophoblasts and multinucleated syncytiotrophoblasts [33]. During syncytialization, cytotrophoblast cells fuse and form an overlying syncytiotrophoblast. Syncytiotrophoblasts arise during syncytialization from the underlying cytotrophoblast cells and are supplied with nutrients through cytoplasmic protrusions from cytotrophoblasts; hence, they are more resistant to pathogens [34]. As the integrity of the placental barrier changes throughout pregnancy, so does the susceptibility to infection—viral passage is easier during early pregnancy, when the syncytiotrophoblast is yet to be fully fused, and before delivery, when the syncytium begins to deteriorate [35]. The infection itself, via mechanical damage or high virus load, can compromise the integrity of the placental barrier [36]. Local factors may also influence the severity of pathological changes caused by infection. For instance, SARS-CoV-2 downregulates the vascular endothelial growth factor (VEGF) signaling, facilitating local ischemia that may manifest as maternal and fetal vascular malperfusion or abnormal placental vessel maturation [37,38]. Recently, vimentin was shown to mediate the entry of the SARS-CoV-2 virus into the cells [39]; therefore, its high levels in the COVID-19 group may indicate placentas’ susceptibility to SARS-CoV-2 infection (Table 5).

### 4.2. SARS-CoV-2 Causes Histopathological Changes in Placenta

Pregnant women infected by SARS-CoV-2 may suffer the consequences of placental disorders even in the absence of macroscopic placental pathologies or symptoms of the infection [40,41,42]. There are no histological changes that are pathognomonic of SARS-CoV-2 infection. Nevertheless, Schwartz et al. defined SARS-CoV-2 placentitis as the coexistence of histiocytic intervillositis, perivillous fibrin deposition, and trophoblast necrosis, which can occur even without confirmed transplacental infection [43,44,45]. The most common histopathological changes in COVID-affected placentas include fetal and maternal vascular malperfusion (FVM and MVM), accelerated villous maturation, massive perivillous fibrin depositions, intervillous thrombi, decidual arteriopathy, and fibrinoid necrosis. Inflammation, such as chronic histiocytic intervillositis, also occurs frequently [30,46]. Importantly, the components of MVM and FVM can occur independently, but the formal diagnosis requires a given set of findings. In some cases, a single extremely prominent feature is sufficient to make or suggest a diagnosis [47].

Emerging evidence suggests that COVID causes significant remodeling of placental vasculature [42]. The placental arteries of COVID-19-positive women had a 5-fold smaller lumen area and 2-fold thicker placental artery walls compared to healthy placentas. Those changes were associated with increased smooth muscle cell proliferation and vascular wall fibrosis. In the study of Gychka et al., such changes occurred in all women who tested positive for COVID-19, regardless of symptoms [42]. The arterial wall thickening and lumen narrowing are expected to alter the blood floor in the placenta. The increase in vascular resistance can drastically decrease blood flow in the placenta, induce a compensatory increase in local blood pressure, and cause local vasculature injury, diminishing the functionality of the placenta [48].

Those findings are reflected in our study since low levels of CD34+ in placentas infected with SARS-CoV-2 may result from their consumption during vascular regeneration, and CD34+ deficiency impairs convalescence after vascular injury (Table 4) [49,50]. Interestingly, Lo et al. showed that the loss of vascular CD34+ results in increased sensitivity to lung injury, suggesting that the CD34 depletion is not specific to the placenta but indicates vascular injury in general [51]. Furthermore, we found moderate or even more severe blood extravasation in most samples of the COVID-19 group, whereas, in the control group, most specimens showed only slight extravasations. That evidence supports the thesis that COVID-19 causes vascular injury and may increase the severity of pathomorphological changes instead of their frequency.

### 4.3. SARS-CoV-2-Associated Intervillositis

Besides the direct cytotoxic effects on host cells, SARS-CoV-2-driven inflammation may also play a crucial role in placental injury. Recent studies showed that macrophage infiltration is a critical factor in contributing to necrotic changes in the affected tissue [52]. Those reports align with our findings, where the expression of macrophage and T cell markers (CD68 and CD4, respectively) were upregulated in the COVID-19 group. Since CD4 and CD68 are usually present on the macrophage surface, macrophages appear to be the driving force of COVID-19-associated inflammation, especially if trophoblast necrosis is present. This rare phenomenon, called chronic histiocytic intervillositis (CHI), is characterized by macrophage infiltration of the intervillous space and fibrin depositions. CHI is associated with an increased risk of fetal growth restriction, miscarriage, and stillbirth. CHI’s etiology remains unclear but is likely associated with autoimmune disorders [53]. Interestingly, CHI may increase the rate of COVID-19 vertical transmission if it coexists with trophoblast necrosis [54,55]. CHI is more common in pregnant women with symptomatic COVID-19, indicating that CHI not only facilitates COVID-19 vertical transmission but is itself a symptom of SARS-CoV-2 infection [56].

Histopathological changes in the placenta may also be associated with chronic inflammation caused by comorbidities or undetected infection during pregnancy. Infarction and fibrin deposition may occur due to autoimmune disease or diabetes mellitus; calcification, while a marker of viral infections, is commonly found in women with pregnancy-induced hypertension or smoking; and maternal and fetal vascular malperfusion is associated with vascular remodeling, which may occur due to thrombosis or hematomas [57,58,59]. The occurrence of undiagnosed viral infection during pregnancy is unlikely in both the research and control group since we found no signs of intrauterine infections or villitis of unknown etiology in any of the patients (Table 3 and Table 4). Since we recruited the patients at random, the presence of comorbidities is likely to affect both groups to the same degree. However, patients with comorbidities may be prone to SARS-CoV-2 infection, leading to a more severe clinical course and the accumulation of histopathological changes in the placenta [60].

In this study, we established several associations between placental pathologies and SARS-CoV-2 infection, including an increased rate of trophoblast necrosis, calcification, intervillositis, and the collapse of intervillous space. Interestingly, our findings show a strong correlation between COVID-19, trophoblast necrosis, and calcification in the placenta tissue, implying uncontrolled cell deaths. Despite the reports about the possible correlation of CHI, trophoblast necrosis, and vertical transmission, none of our tested newborns presented positive COVID-19 PCR examinations. However, due to the relatively small size of the tested group and the rarity of SARS-CoV-2 vertical transmission, it is impossible to draw clear conclusions.

## 5. Limitations

This research cohort was collected during the early months of the pandemic when the serotypes alpha and delta of SARS-CoV-2 were the most prevalent. Therefore, the described pathological changes are characteristic of those serotypes and may not reflect the virulence of milder serotypes. We cannot rule out that the slight difference in the gestation age of patients from both cohorts at the time of delivery had an impact on our findings. However, since patients were recruited randomly, patients with SARS-CoV-2 infection were otherwise a healthy cohort without prior medical history of infections—in this cohort, SARS-CoV-2 was the main factor associated with preterm birth. On the other hand, preterm birth is a rare occurrence in healthy patients, which constituted the control group. While our group consists only of patients with active SARS-CoV-2 infection during delivery, the relatively small cohort size may make the generalization of our conclusion difficult. Still, our group consists only of patients with active SARS-CoV-2 during delivery. Finally, the findings may be influenced by the regional specificity of SARS-CoV-2 and not reflect the general population of pregnant women.

## 6. Conclusions

SARS-CoV-2 is associated with various pathological changes in the placenta. In our cohort, intervillositis, calcifications, the collapse of intervillous space, and trophoblast necrosis were more common and severe in COVID-affected placentas than in healthy ones. The changes found, such as calcification, form during viral infection and may reflect placental insufficiency. The expression levels of CD4+, a T cell marker, and CD68+, a macrophage marker, were significantly higher in the COVID-19 group than in control samples, suggesting that T lymphocytes and macrophages drive the pathological changes associated with SARS-CoV-2 infection. The expression of CD34+, representative of the vascular endothelium, was lower in the COVID-19 group, suggesting the presence of blood vessel injuries related to SARS-CoV-2. We suspect that the accumulation of CD4+/CD68+ reflects histiocytic intervillositis in placental tissue, which facilitates placental injury and may contribute to pregnancy complications. The coexistence of accelerated vascular maturation and trophoblast necrosis suggests that SARS-CoV-2 may impact the placenta not only directly by destroying trophoblast cells but also indirectly by interfering with the placental vasculature. While new reports regarding the mechanisms of SARS-CoV-2-associated placentitis and the remodeling of placental vasculature are emerging, how they impact pregnancy outcomes is still up for debate.

## Figures and Tables

**Figure 1 diseases-12-00142-f001:**
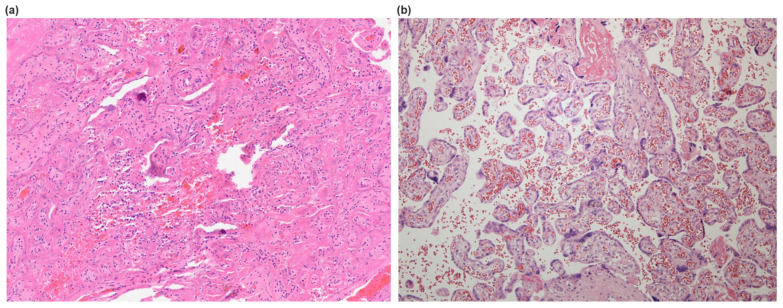
Representative cross-sectional staining patterns at ×10 magnitude of (**a**) COVID-19 placenta with blood extravasation, the collapse of intervillous space, calcification, and fibrin depositions and (**b**) healthy placenta.

**Figure 2 diseases-12-00142-f002:**
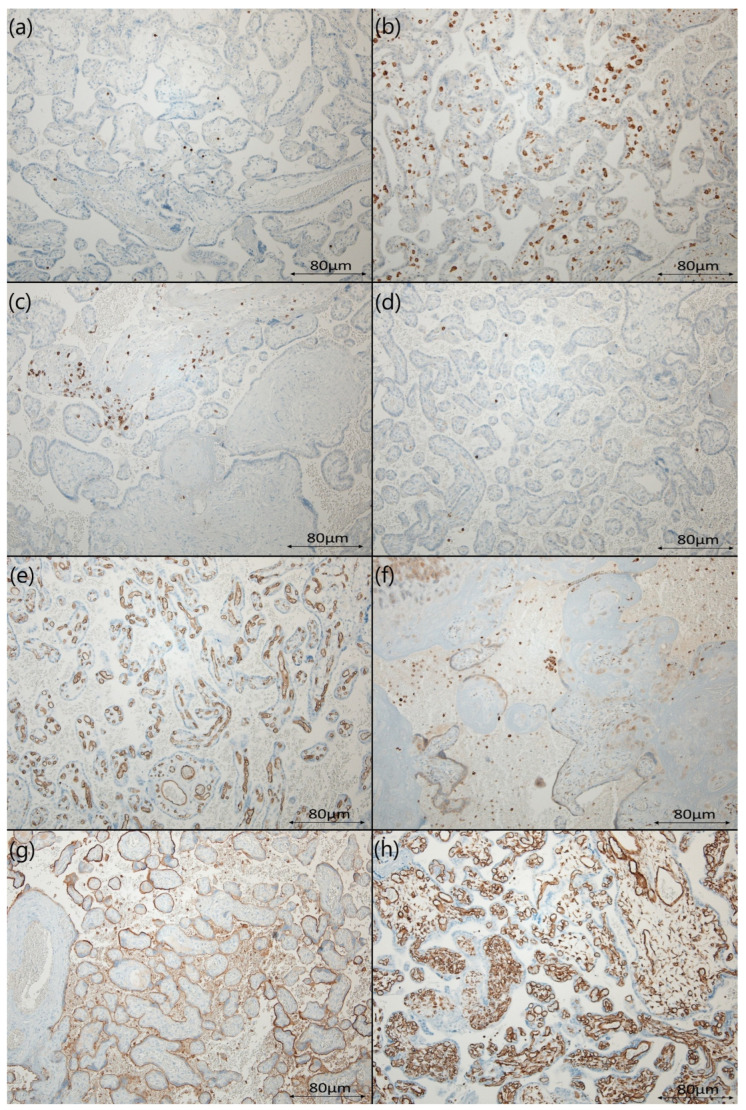
Representative cross-section staining patterns of placental tissues stained with (**a**) anti-CD3 antibodies (T cells); (**b**) anti-CD4 antibodies (T cells); (**c**) anti-CD8 antibodies (cytotoxic T cells); (**d**) anti-CD20 antibodies (B cells); (**e**) anti-CD34 antibodies (vascular epithelium); (**f**) anti-CD68 antibodies (macrophages); (**g**) anti-CD138 antibodies (plasmacytes); (**h**) anti-vimentin antibodies. In each image, cells expressing IHC markers are stained brown.

**Table 1 diseases-12-00142-t001:** Characteristics of antibodies used for immunohistochemical analysis.

Antibody	Product ID	Description	Location/Control	Staining
CONFIRM anti-CD3 (2GV6)	05278422001, 790-4341	Rabbit, primary, monoclonal	Cytoplasm, Membrane/Tonsil	16 min, 37 °C
CONFIRM anti-CD4 (SP35)	05552737001, 790-4423	Rabbit, primary, monoclonal	Membrane/Tonsil	16 min, 37 °C
CONFIRM anti-CD8 (SP57)	05937248001, 790-4460	Rabbit, primary, monoclonal	Membrane/Tonsil	16 min, 37 °C
CONFIRM anti-CD20 (L26)	05267099001, 760-2531	Mouse, primary, monoclonal	Membrane/Tonsil	8 min, 37 °C
CONFIRM anti-CD34 (QBEnd/10)	05278210001, 790-2927	Mouse, primary, monoclonal	Membrane/Tonsil	12 min, 37 °C
CONFIRM anti-CD68 (KP-1)	05278252001, 790-2931	Mouse, primary, monoclonal	Membrane/Tonsil	16 min, 37 °C
anti-CD138 (B-A38), PAb, Cell Marque	05269083001, 760-4248	Mouse, primary, monoclonal	Membrane/Tonsil	30 min, 37 °C
CONFIRM anti-Vimentin (V9)	05278139001, 790-2917	Mouse, primary, monoclonal	Cytoplasm/Tonsil	16 min, 37 °C

**Table 2 diseases-12-00142-t002:** Patient characteristics.

Variable	COVID-19 Group (n = 18 *)	Control (n = 22)
Age	31 (range 24–38)	31 (range 22–43)
Gestational age at delivery	34 (range 24–38)	38 (range 31–41)
Gestational age at COVID-19 diagnosis	34 (range 24–38)	-
Mean newborn body mass	2140 g (range 644–2980 g)	2395 g (1200–3680 g)
Apgar ≥ 7 (1’)	5/18 (35.71%)	19/22 (86.36%)
Cough	4/18 (22.22%)	0/22 (0%)
Dyspnea	9/18 (50%)	0/22 (0%)
Placenta with positive PCR test	13/15 ** (86.67%)	0/22 (0%)

Data presented as mean. * The difference between COVID-19 groups results from lacking clinical data in five cases. The histopathological data were preserved and are presented in the table. ** In 8 cases, the test was highly suggestive of positive COVID-19, but it was deemed undiagnostic after special consideration.

**Table 3 diseases-12-00142-t003:** Comparison of histological findings in placentas of COVID-19-positive and healthy women based on the Amsterdam criteria.

Placental Lesion	Pathology	Occurrence/ Intensity	COVID-19 Group (n = 23)	Control Group (n = 22)	*p*-Value
	Infarction	Negative	18	16	*p* = 0.43
		Positive	3	5	
	Retroplacental hemorrhage	Negative	9	6	*p* = 0.99
		Positive	0	0	
Maternal vascular malperfusion	Distal villous hypoplasia	Negative	4	10	*p* = 0.062
		Positive	16	11	
	Accelerated vascular maturation	Negative	1	7	***p* = 0.022** More in COVID-19 group
		Positive	19	14	
	Decidual arteriopathy	Negative	0	1	*p* = 0.36
		Positive	14	12	
Maternal vascular malperfusion (total)		Negative	1	3	*p* = 0.22
		Positive	20	15	
	Venous thrombosis	Negative	12	13	*p* = 0.75
		Positive	9	8	
	Arterial thrombosis	Negative	16	18	*p* = 0.43
		Positive	5	3	
Fetal vascular malperfusion	Avascular villi	Negative	3	4	*p* = 0.73
		Positive	17	17	
	Intramural fibrin depositions	Negative	13	15	*p* = 0.84
		Positive	6	6	
	Stroma obliteration	Negative	3	6	*p* = 0.35
		Positive	17	15	
	Vascular ectasia	Negative	7	5	*p* = 0.44
		Positive	13	16	
Fetal vascular malperfusion (total)		Negative	2	6	*p* = 0.109
		Positive	16	12	
Delayed vascular maturation		Negative	13	13	*p* = 0.67
		Positive	6	8	
Villitis of unknown etiology		Negative	20	20	*p* = 0.99
		Positive	0	0	

Results that are statistically significant are shown in bold.

**Table 4 diseases-12-00142-t004:** Other histological findings in COVID-19-affected and healthy placenta.

Features	Occurrence/ Intensity	COVID-19 Group (n = 23)	Control Group (n = 22)	*p*-Value
Diffuse fibrin depositions	Sparse	8	13	*p* = 0.30
	Intermediate	9	8	
	Rich	6	1	
	Positive	6	6	
Calcifications	No	4	17	***p* = 0.0004** More in COVID-19
	Yes	19	5	
Intervillous space collapse	No	5	11	***p* = 0.015** More in COVID-19
	Yes	18	11	
Intervillositis (intensity)	Slight	9	20	***p* = 0.001** More severe in COVID-19
	Moderate	8	2	
	Considerable	2	0	
	Severe	4	0	
Blood extravasations	Slight	1	17	*p* = 0.4
	Moderate	13	5	
	Considerable	6	0	
	Severe	2	0	
	Unknown	1	0	
Trophoblast necrosis	No	3	22	***p* < 0.001** More in COVID-19
	Yes	20	0	
Intrauterine infection	Negative	20	21	*p* = 0.99
	Positive	0	0	

Results that are statistically significant are shown in bold.

**Table 5 diseases-12-00142-t005:** The difference in immunohistochemical expression between samples from COVID-19-positive patients and healthy women.

Variable	Intensity	COVID Group (n = 23)	Control Group (n = 22)	Intergroup Difference
CD3+	0	1	6	*p* = 0.07
	1	14	13	
	2	6	0	
	3	2	0	
	Unknown *	0	3	
CD4+	1	1	0	***p* = 0.01** Higher in COVID-19
	2	10	19	
	3	6	3	
	4	6	0	
CD8+	1	17	14	*p* = 0.33
	2	5	8	
	Unknown *	1	1	
CD20+	1	21	18	*p* = 0.35
	2	2	4	
CD34+	Low	18	11	***p* = 0.048** Lower in COVID-19
	High	5	11	
CD68+	1	1	1	***p* = 0.0006** Higher in COVID-19
	2	10	20	
	3	7	0	
	4	4	0	
	5	1	0	
	Unknown *	0	1	
CD138+	1	23	18	*p* = 0.27
	2	0	1	
	Unknown *	0	3	
Vimentin	Low	13	20	***p* = 0.009** Higher in COVID-19
	High	10	2	

Results that are statistically significant are shown in bold. Here, 0 represents a lack of positive cells; 1—<1%; 2—1–5%; 3—5–10%; 4—>10–25%; 5—>25%. *—the tissue sample was nondiagnostic.

**Table 6 diseases-12-00142-t006:** Statistically significant and borderline significant correlations between histopathological changes and immunohistochemical markers in placentas of COVID-19-positive women (n = 23).

Histological Change	Immunohistochemistry	*p*-Value	Correlation Coefficient (k)
Diffuse fibrin depositions	CD8	0.026	k = 0.49
	CD20	0.0514	k = 0.42
	CD68	0.014	k = 0.35
Calcifications	CD34	0.043	k = −0.48
Intervillositis (intensity)	CD20	0.074	k = 0.39
	CD68	0.053	k = 0.516
Blood extravasations	CD8	0.076	k = 0.36
	CD68	0.0961	k = 0.32
Delayed vascular maturation	CD3	0.046	k = −0.48
Infarction	CD34	0.064	k = 0.46
	CD20	0.02	k = 0.55

## Data Availability

The data presented in this study are available on request from the corresponding author. However, due to ethical restrictions, the data are not publicly available.

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
