# Peer review of "SARS-CoV-2 Infection during Delivery Causes Histopathological Changes in the Placenta"

_diseases, 2024, doi:10.3390/diseases12070142_

Round 1
Reviewer 1 Report
Comments and Suggestions for Authors
Line 37- conclusion should be according to your results (eg results of our study suggest…)
Line 49- define abbreviation on first mentioning (ARDS)
Line 63- why is font smaller
Line 86- please add study setting (town, country)
Line 135- please add reference for your protocol
Line 213- add under the table the data are presented as mean/median (standard deviation…)
Line 244- please use . For decimal numbers (not ,)
Line 358- again change in font size
Please add limitation section
Overall, this is interesting and valuable study. English language and scientific style could be improved throughout the manuscript. I believe this study adds to the body of literature in the field of COVID effects on pregnancy. Tables and figures are nicely presented and literature is up to date. I believe conclusion could be rewritten in order for it to align with the results of the study.
Comments on the Quality of English LanguageShould be improved
Author Response
Dear Reviewer, thank you for reviewing our work! We revised our manuscript and updated it including your suggestions. Please, find our responses below.
S1: Line 37- conclusion should be according to your results (eg results of our study suggest…)
R: Conclusions have been rewritten and updated (in abstract and in the manuscript).
S2: Line 49- define abbreviation on first mentioning (ARDS)
R: We defined the abbreviation on first mentioning.
S3: Line 63- why is font smaller
R: Font has been updated.
S4: Line 86- please add study setting (town, country)
R: We added this data to the manuscript.
S5: Line 135- please add reference for your protocol
R: Since our protocol has not been published and is currently unavailable for the outside use, we can provide a direct reference. However, it was adapted for use following the “Tissue pathway for histopathological examination of the placenta” devised by The Royal College of Pathologists (Evans et al. 2022).
S6: Line 213- add under the table the data are presented as mean/median (standard deviation…)
R: Explained that the data are expressed as mean.
S7: Line 244- please use . For decimal numbers (not ,)
R: We revised the whole manuscript and changes all , to . for decimal numbers.
S8: Line 358- again change in font size
R: We revised the manuscript and updated the font.
S9: Please add limitation section
R: “Limitations” has been added to the manuscript.
S10: Overall, this is interesting and valuable study. English language and scientific style could be improved throughout the manuscript. I believe this study adds to the body of literature in the field of COVID effects on pregnancy. Tables and figures are nicely presented and literature is up to date. I believe conclusion could be rewritten in order for it to align with the results of the study.
R: Thank you! We revised the language and rewrote the conclusions.
Reviewer 2 Report
Comments and Suggestions for Authors
In this manuscript authors investigated the histopathological changes found in COVID-19-affected placentas and found that COVID-19 placentas showed fibrin depositions, calcifications and inflammation. Moreover, trophoblast necrosis was found only in the placentas of the research group. Moreover, authors found an increased expression of CD68+ in the COVID-19 cohort.
The topic is interesting but the manuscript needs important revisions. See my comments below.
Line 49-50: It deserves to be highlighted that COVID-19 can also lead to pregnancy and fertility complications (see PMID: 35114008 and PMID: 35943095)
2.6. Immunohistochemical staining: The primary antibodies must be reported with the relative product code and should be moved in a dedicate table
Table 1. Patient characteristics: Only age-matched samples can be compared. Adjust the sample size according to the gestation age at delivery. From what I see there are 10 Gynaecologists among the authors. It should not be difficult to collect appropriate tissues.
Figure 1: Higher magnifications and scale bars are necessary
Table 4: Representative IHC images for each CD marker tested must be added
An accurate revision of typing errors and formatting are necessary.
Syntax and grammar must be reviewed
Abbreviations must be written in full length when mentioned for the first time
It is surprising that such a simple study required 17 authors. Moreover, from the authors contribution section I see that 3 authors wrote the paper and 6 authors "conceived and designed the analysis" or "supervised the project". However, no equally contributed are present. A clear and detailed author contribution section must be added.
Comments on the Quality of English LanguageSyntax and grammar must be reviewed
Author Response
Dear Reviewer, thank you for time and effort in reviewing our work! We updated the manuscript with consideration to your suggestions. Please, find our responses below.
S1: Line 49-50: It deserves to be highlighted that COVID-19 can also lead to pregnancy and fertility complications (see PMID: 35114008 and PMID: 35943095)
R: Thank you for the suggestion! We added the suggested article (PMID: 35943095) to the manuscript and explained that SARS-CoV-2 predisposes placenta injury and may be interlinked with preeclampsia. We also included data about the possible risk of infertility after SARS-CoV-2 infection.
S2: 2.6. Immunohistochemical staining: The primary antibodies must be reported with the relative product code and should be moved in a dedicate table
R: A table description all used antibodies has been added.
S3: Table 1. Patient characteristics: Only age-matched samples can be compared. Adjust the sample size according to the gestation age at delivery. From what I see there are 10 Gynaecologists among the authors. It should not be difficult to collect appropriate tissues.
R: This specific group was collected during the early months of the pandemic, when the serotypes alpha and delta were the most prevalent. At this time, it was not widely recognised that SARS-CoV-2 is associated with preterm birth, which is reflected by the differences in the mean gestational age of our patients. Importantly, the patients were recruited at random. Patients with SARS-CoV-2 infection were a healthy cohort, without prior medical history of infections – in this cohort SARS-CoV-2 was the main factor associating with preterm birth. However, preterm birth is a rare occurrence in healthy patients, which constituted the control group.
S4: Figure 1: Higher magnifications and scale bars are necessary
R: The Figure was split into two parts: one includes comparative healthy and SARS-CoV-2-affected placenta, the other shows the patterns of used IHC stainings. The quality of the used images was improved and scale bars were added.
S5:Table 4: Representative IHC images for each CD marker tested must be added
R: A figure that shows shows the patterns of all used IHC stainings was added.
S6: An accurate revision of typing errors and formatting are necessary. Syntax and grammar must be reviewed
R: We revised the manuscript to improve grammar and style.
S7: Abbreviations must be written in full length when mentioned for the first time
R: We explained all abbreviations on first mention.
S8: It is surprising that such a simple study required 17 authors. Moreover, from the authors contribution section I see that 3 authors wrote the paper and 6 authors "conceived and designed the analysis" or "supervised the project". However, no equally contributed are present. A clear and detailed author contribution section must be added.
R: Authors contribution statements were revised and a AC form was provided as an additional PDF along the resubmitted manuscript.
Reviewer 3 Report
Comments and Suggestions for Authors
Dear Author/s,
Manuscript ID: diseases-3056574 “Histological changes and their implications in COVID-19-affected placentas” contributes valuable insights into the prevalence and factors contributing to changes in placentas of infected patients. The manuscript's comprehensive data collection and statistical analysis provide a strong foundation for its conclusions. Addressing the suggested improvements and revisions would enhance the manuscript's impact and readability
Suggestions for Improvement:
1. Relevance and Significance: The study addresses an important topic, particularly in the context of the COVID-19 pandemic. The manuscript's emphasis on histopathological analysis derived from COVID-19 infected patients is relevant and its findings contribute to our understanding of the factors that impacted in childbirth. However, sample cohort is very low to establish any finding.
2. Literature Review and Context: The manuscript could benefit from a more comprehensive review of the existing literature on histopathological dynamics, especially in the context of different other variants prevalent during pandemics. This would help the reader understand how the current study contributes to the field and differentiates itself from previous research with other viral infections.
3. Limitations and Generalizability: While the manuscript acknowledges some limitations, such as the lack of other data derived from clinical sample, and the potential for regional bias, further discussion on the generalizability of the findings to other contexts would enhance the manuscript's completeness.
4. Too much elaboration of covid diagnosis method is not necessary, PCR testing etc. Citing papers like PMID: 36016088 could be useful and solve the purpose (section 2.3).
Minor Revisions:
1. The title of the paper seems incomplete and doesn’t look good.
2. Inclusion of the scale bar in the figure would be more convincing.
3. The figures and tables provided are essential for understanding the study's results. To enhance clarity, consider providing more descriptive statistics on the table and figure captions that explain the main findings depicted in each figure.
I would recommend justifying all these minor mistakes at your end. Above are some examples; authors should take care of rest likewise
All the best.
The manuscript's language and writing style are generally clear and concise. However, some sentences appear to be lengthy and could be restructured for better readability. In the method and discussion section language correction is needed. Authors should proof-read (English and Grammar) the manuscript at your end.
Author Response
Dear Reviewer, thank you for your time and effort in reviewing our work! Please, find our revised manuscript and responses to your suggestions below.
Suggestions for Improvement:
S1: Relevance and Significance: The study addresses an important topic, particularly in the context of the COVID-19 pandemic. The manuscript's emphasis on histopathological analysis derived from COVID-19 infected patients is relevant and its findings contribute to our understanding of the factors that impacted in childbirth. However, sample cohort is very low to establish any finding.
R: The significance of the topic was updated in the introduction and an additional section “Limitations” was added to the manuscript. We acknowledge that our cohort is relatively small, but this specific group was collected during the early months of the pandemic, when the serotypes alpha and delta were the most prevalent. At this time, it was not widely recognised that SARS-CoV-2 is associated with preterm birth, which is reflected by the differences in the mean gestational age of our patients. Importantly, the patients were recruited at random. Patients with SARS-CoV-2 infection were a healthy cohort, without prior medical history of infections – in this cohort SARS-CoV-2 was the main factor associating with preterm birth. However, preterm birth is a rare occurrence in healthy patients, which constituted the control group.
S2: Literature Review and Context: The manuscript could benefit from a more comprehensive review of the existing literature on histopathological dynamics, especially in the context of different other variants prevalent during pandemics. This would help the reader understand how the current study contributes to the field and differentiates itself from previous research with other viral infections.
R: We added the description of the specificity of our cohort to the manuscript. We also revised the discussion to include possible pathomechanism in which pathological changes in placenta develop.
S3: Limitations and Generalizability: While the manuscript acknowledges some limitations, such as the lack of other data derived from clinical sample, and the potential for regional bias, further discussion on the generalizability of the findings to other contexts would enhance the manuscript's completeness.
R: We added an additional “Limitations” section.
S4: Too much elaboration of covid diagnosis method is not necessary, PCR testing etc. Citing papers like PMID: 36016088 could be useful and solve the purpose (section 2.3).
R: The methodology was updated and shortened. Appropriate citations were added.
S5: The title of the paper seems incomplete and doesn’t look good.
R: The title was changes to better reflect the content of the manuscript.
S6: Inclusion of the scale bar in the figure would be more convincing.
R: We added scale bars to the figures.
S7: The figures and tables provided are essential for understanding the study's results. To enhance clarity, consider providing more descriptive statistics on the table and figure captions that explain the main findings depicted in each figure.
R: We added descriptions under tables and figures.
Round 2
Reviewer 2 Report
Comments and Suggestions for Authors
the manuscript has been improved and can be accepted in the current form
Author Response
Dear Reviewer,
thank you for your time and effert in reviewing our manuscript! We checked it once again to ensure its quality.
Reviewer 3 Report
Comments and Suggestions for Authors
I would like to appreciate the efforts of authors for making the necessary amendments to the manuscript (Diseases-3056574-v2) and provide reasonable answers to the queries raised by the reviewers. It is good to have the appropriate references and/or detailed methods, especially when you are using any establish or novel methods.
The current form of the manuscript is looks much better now, although I can see there some minor amendment still needed, like incorporate the catalogue numbers of the molecular biology or staining products etc. I would like to urge please take care of these minor things at your end. Proofread the manuscript one more time and try to make it flawless as much as you can.
All the best.
Comments on the Quality of English LanguageI would like to urge please proofread the manuscript one more time and try to make it flawless as much as you can.
Author Response
Dear Reviewer, we really appreciate your help in improving our manuscript! We added the catalogue numbers to the table and methodology. The manuscript was also once more revised and proofread to fix all minor mistakes on our side.